# Controlling *Fusarium oxysporum* Tomato Fruit Rot under Tropical Condition Using Both Chitosan and Vanillin

Zahir Shah Safari [ORCID], Phebe Ding *, Jaafar Juju Nakasha and Siti Fairuz Yusoff [ORCID]

Department of Crop Science, Faculty of Agriculture, Universiti Putra Malaysia, Serdang 43400, Malaysia; zahirshah.safari@gmail.com (Z.S.S.); jujunakasha@upm.edu.my (J.J.N.); yuezyusoff@gmail.com (S.F.Y.)
* Correspondence: phebe@upm.edu.my

**Abstract:** Tomato *Lycopersicon esculentum* Mill. is one of the most cultivated and widely consumed vegetables in the world. However, it is very susceptible to the infection initiated by *Fusarium oxysporum* fruit rot, which shortens post-harvest life and thus reduces market value. This disease can be regulated appropriately by the application of synthetic fungicides. However, chemical fungicides constitute a serious health risk, and have harmful environment effects and increase disease resistance, even when microbes are dead. Hence, to overcome this problem, chitosan and vanillin, which have antimicrobial bioactive properties against the growth of microorganisms, could be an alternative to disease control, while maintaining fruit quality and prolonging shelf life. The aim of this research was to evaluate the antimicrobial activity of chitosan and vanillin towards the inoculate pathogen and to investigate the effect of chitosan and vanillin coating in vivo on *Fusarium oxysporum* fruit rot and defense-related enzymes (PAL, PPO and POD). Chitosan and vanillin in aqueous solutions, i.e., 0.5% chitosan + 10 mM vanillin, 1% chitosan + 10 mM vanillin, 1.5% chitosan + 10 mM vanillin, 0.5% chitosan + 15 mM vanillin, 1% chitosan + 15 mM vanillin and 1.5% chitosan + 15 mM vanillin, were used as edible coatings on tomatoes stored at $26 \pm 2\,°C$ and $60 \pm 5$ relative humidity. The result revealed 1.5% chitosan + 15 mM vanillin was able to control disease incidence by 70.84% and severity by 70%. These combinations of coatings were also able to retain phenylalanine ammonia-lyase (PAL), peroxidase activity (POD), and polyphenol oxidase (PPO) enzyme activities as well as prolong shelf life of tomatoes up to 15 days.

**Keywords:** postharvest disease; antioxidant activity; postharvest losses; protein; phenylalanine ammonia-lyase (PAL); peroxidase activity (POD); polyphenol oxidases (PPO); Fusarium fruit rot

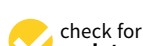

## 1. Introduction

Tomato fruit (*Solanum lycopersicum* Mill.) is the second most important vegetable after the potato, and it can be consumed either raw or cooked [1]. Tomato fruit contains impressive health benefits due to its significant bioactive antioxidant compounds with β-carotene, lycopene, flavonoids, organic acids, chlorophyll and vitamins [2]. In addition, it is also low in fat and calories. Tomato fruit is a very delicate fruit vegetable that is susceptible to high postharvest losses, which vary from country to country. The postharvest loss of tomato fruit was 17.26% in India, 12% in the US, 13.75% in Egypt and 26% in Bangladesh [3,4]. One of the prominent losses was due to pathological damage that affects the quality and nutritional value of this fruit [5]. Moreover, pathological damage causes mycotoxin contamination and market value reduction of the fresh produce. Fusarium fruit rot caused by *Fusarium oxysporum* is a common disease in tomato fruit and is capable of causing fruit spoilage either in the field or in storage [6]. The application of synthetic fungicides such as carbendazim, chlorothalonil and mancozeb [7] is commonly used to control this disease. However, this control measure has negative impacts on human health and the environment due to chemical residues and an increase in pathogen resistance [8]. Furthermore, public anxiety over sustainable food production and safety has resulted in

attempts to find new control agents for postharvest diseases; this has led us to study the effects of combining chitosan and vanillin as natural antimicrobials in controlling tomato postharvest disease.

Chitosan (poly-β-(1-4)-*N*-acetyl-D-glucosamine) and its derivative have been reported to be a potent and biodegradable alternative to synthetic fungicides [9]. Earlier studies indicated that chitosan was effective in controlling postharvest decay of many fruits and vegetables, including pomegranates, tomato fruit, strawberries, potatoes, table grapes, apples and peaches [10,11]. In addition, vanillin (4-hydroxy-3-methoxy benzaldehyde), a phenolic aldehyde organic compound derived from the vanilla bean [12], has been proven to control *Alternaria alternata*, which causes *Alternaria* rot disease in table grapes during storage at $4 \pm 2\,^{\circ}$C [8]. Temperature management during storage is also an important factor in disease management, fruit quality and the extension of shelf life. However, in developing countries, most growers and retailers store tomatoes at ambient conditions ($26 \pm 2\,^{\circ}$C) due to a lack of cool storage facilities [13]. Nevertheless the effects of a coating prepared by combining chitosan, which has quality maintenance properties, and vanillin, which has antimicrobial properties, in controlling Fusarium fruit rot and postharvest quality of tomatoes at $26 \pm 2\,^{\circ}$C has yet to be studied. Therefore, this study was conducted to determine the potential of chitosan in combination with vanillin as a coating in controlling tomato fruit Fusarium fruit rot.

## 2. Materials and Methods

### 2.1. Fruit Materials

Pink color tomato fruit (10% to 30% of the surface is yellow to pink, according to USDA class 3 color) from the Syngenta 1039 variety were obtained from Weng Seng Vegetable Products Sdn. Bhd., Pahang, Malaysia. On the same day of harvesting, tomato fruit was sent to the Laboratory of Postharvest, Department of Crop Science, Faculty of Agriculture, Universiti Putra Malaysia. Fruit free from any defects and diseases, with uniform shape, maturity, color and weight and ranging between 90–110 g was used in this study.

### 2.2. Pathogen Inoculation

*Fusarium oxysporum* (MT012284) were originally isolated from tomato fruit showing fruit rot symptoms; the outer surface of the infected fruit appeared as a pale white lesion, with powdery discolored spots covered by white and pinkish mycelium. The infected area was softer and slightly sunken as compared to unaffected fruit parts (Figure 1).

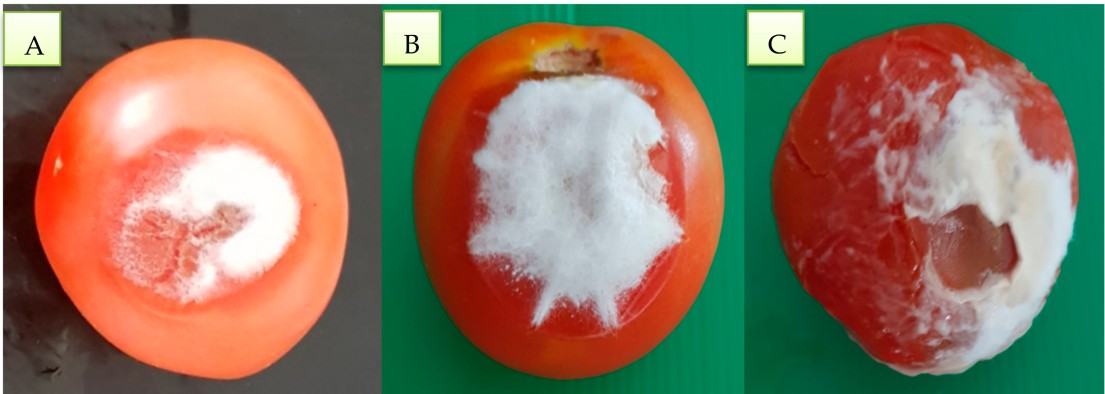

**Figure 1.** Symptoms of tomato fruit rot: soft and sunken tissue covered by white pinkish mycelium. **A**—20% severity, **B**—40% severity, and **C**—more than 75% severity.

*Fusarium oxysporum* were identified based on their morphological cultural traits on fungus colony culture (conidia shape), and morphological identification was confirmed by molecular identification. Fungal DNA was extracted from freshly collected mycelium of 7-day-old cultures using the DNA extraction Kit (QIAGEN DNA Mini

Kit, HB-1166, Hilden, Germany). The internal transcribed spacer (ITS) region of rDNA was amplified using the universal primers ITS1 and ITS4 (Kulatunga et al., 2016). The forward primer ITS1 (5′-TCCGTAGGTGAACCTGCGG-3′) and reverse primer ITS4 (5′-TCCTCCGCTTATTGATATGC-3′) were synthesized by First Base Laboratories Sdn Bhd, Malaysia. The ITS sequence was compared with the NCBI GenBank database (www.ncbi.nlm.nih.gov/blast, accessed on 17 May 2019) sequences using the BLAST search tool. The analyses supported the results obtained in the morphological study and thus confirmed that *Fusarium oxysporum* was identified as the causative agent of tomato fruit rot.

Tomato fruit rot causative pathogen *Fusarium oxysporium* was cultured and incubated for 7 days at $26 \pm 2$ °C and $60 \pm 5$% RH. Sterilized distilled water of 10 mL was poured into 1-week-old *Fusarium oxysporium* culture, and the surface was scraped lightly with a bent glass rod. The obtained conidial suspension was filtered over a double-layer sterilized muslin cloth and centrifuged for 5 min at $4000 \times g$. The conidial counts were adjusted to $2 \times 10^6$ conidia per mL using a hemocytometer. The surface of tomato fruit was sterilized for 3 min in 0.05% sodium hypochlorite, then washed using tap water and air-dried under sterile conditions for 2 h. After drying, the fruit was dipped into antagonistic conidial suspension for 1 min, and then the fruit was allowed to dry at $26 \pm 2$ °C and $60 \pm 5$% RH for 2 h.

### 2.3. Preparation of Coating Solutions

The chitosan originated from shrimp-shell crustaceans with 85% deacetylation, purchased from Enviro Clean Energy Sdn. Bhd. Perintis Teknologi Pertanian, Malaysia. Meanwhile, an organic compound of 99% pure vanillin with the molecular formula $C_8H_8O_3$ was bought from Evergreen Engineering & Resources Sdn. Bhd., Selangor, Malaysia. Chitosan solutions with concentrations of 0.5%, 1%, and 1.5% *v/v* were prepared, the solution pH was adjusted to 5.6 with 1 M NaOH, and 0.1% Tween 20 (polyoxyethylene sorbitan monooleate, Sigma Aldrich) was added to improve the solution wettability. Distilled water containing 0.1% Tween 20 without chitosan served as a control. Vanillin powders were dissolved in 83 °C distilled water to obtain a 10 and 15 mM concentration solution by heating. Then, each vanillin solution was combined with the three chitosan concentration solutions to form 0.5% chitosan + 10 mM vanillin, 1% chitosan + 10 mM vanillin, 1.5% chitosan + 10 mM vanillin, 0.5% chitosan + 15 mM vanillin, 1% chitosan + 15 mM vanillin and 1.5% chitosan + 15 mM vanillin coating solutions.

### 2.4. Postharvest Coating Treatments

Tomato fruit with chlorinated water prepared from 0.05% sodium hypochlorite was dipped for 3 min prior to coating treatments [14]. The fruit was rinsed and air dried for 1 h and randomly divided into 8 lots. All 8 lots of fruit were dipped for 1 min in the coating solutions. The negative control consisted of fruit without inoculation, while the positive control fruit was inoculated with *Fusarium oxysporum* and then dipped in distilled water containing 0.1% Tween 20 for 1 min. All of the fruit was dried for 2 h at $26 \pm 2$ °C and $60 \pm 5$% RH. For each coating, six fruit per replicate were used. The fruit was then packed in 18 cm × 26 cm plastic bags of 0.05 mm thickness containing 18 holes 0.5 cm in diameter. These bags were placed in commercial corrugated fiberboard cartons of 30 cm × 25 cm × 15 cm. The cartons were then stored at $26 \pm 2$ °C and $60 \pm 5$% RH for 15 days. Each treatment was repeated four times and analysis was carried out at and interval of every 3 days.

### 2.5. Determination of Disease Incidence

The disease incidence (DI) was measured as the percentage of fruit displaying fruit rot symptoms, according to the method of Khaliq [15]. The DI was determined as the number of infested fruit showing symptoms of the disease, such as dots and rots, out of the total number of tomato fruit for each batch and storage interval. Six tomato fruit were

distributed and used for DI. The percentage of disease was determined using the following formula (Equation (1)), as reported by Abebe et al. (2017):

$$\text{DI } (\%) \; = \; \frac{\sum(\text{DI level}) \; \times \; (\text{Number of tomato fruit at the DI level})}{\text{Total number of tomato fruit in the treatment} \; \times \; \text{The highest score (5)}} \times 100\% \qquad (1)$$

### 2.6. Determination of Disease Severity

　　Tomato fruit disease severity (DS) was evaluated as described by Mohamed [16], with slight modification. Fruit DS was evaluated based on visible symptoms, spots, rot, and decayed areas on each fruit surface at every storage interval. For DS assessment, five DS scores were used, as shown in Table 1. Fruit with index scores of two, three, and four were considered to have no commercial and marketing value (Equation (2)).

$$\text{DS } (\%) = \frac{\sum(\text{Severity rating} \times \text{Number of tomato fruit clusters in the rating})}{\text{Total number of tomato fruit clusters assessed} \times \text{Highest DS scale}} \times 100\% \qquad (2)$$

**Table 1.** Disease severity scores for disease assessment of tomato fruit.

| Diseases Score | Description | Inference |
|:---:|:---:|:---:|
| 0 | No visible symptoms on fruit | No infection |
| 1 | 1–25% of the area covered by slight necrotic inoculations | Mild infection |
| 2 | 26–50% of the inoculated area covered by necrotic and white fungal mycelia | Moderate infection |
| 3 | 51–75% of the sample is necrotic with the presence of spore mass | Severe infection |
| 4 | >76% Necrotic tissue with fungal mass; appears soft and decayed | Very severe/Devastating |

### 2.7. Determination of Antioxidant Capacity and Activity

2.7.1. Supernatant Extraction

　　Tomato fruit pulp tissue extraction assays for total phenolic content (TPC) and antioxidant activities Radical scavenging activity by using (2,2-azino-bis (3-ethylbenzthiazoline-6-sulfonic acid) (ABTS), Ferric Reducing Antioxidant Power (FRAP) were extracted using methods defined by Rastegar and Si [17,18], with slight modification. Four grams of tomato fruit tissue from each replicate was frozen immediately in liquid $N_2$ and minced using a small ceramic kitchen pestle and mortar for 30 s. The ground tissue was dissolved in 10 mL 80% (*v/v* methanol analytical grade) and then transferred to a 100 mL conical flask, which was covered with aluminum foil. Subsequently, the homogenate was extracted under reduced light conditions by spinning on an orbital shaker at 180 rpm for 1 h. After shaking, the homogenate was filtered by Whatman No. 1 filter paper and transferred to a vial, which was covered with aluminum foil; the supernatant was kept at −20 °C until analyses.

2.7.2. Determination of Total Phenolic Content

　　The total phenolic content was estimated following the method described by Zainal [19], with some modifications. In brief, 150 µL aliquot of supernatant extract and 750 µL of 10% (*v/v*) Folin Ciocalteu reagent were mixed in test tubes covered with aluminum foil and incubated for 5 min in darkness. This was followed by the addition of 600 µL of 7.5% (*w/v*) sodium carbonate ($Na_2CO_3$). The mixture was then incubated in darkness for 30 min at room temperature 26 ± 2 °C and 60 ± 5% RH, before measuring the absorbance at 765 nm with a spectrophotometer (S1200, Spectrowave spectrophotometer, Cambridge, UK). The total phenolic content was expressed as milligram gallic acid equivalent (GAE) per 100 g

fresh weight (FW), using gallic acid as the standard with $R^2 = 0.97$ and calculated using the following equation (Equation (3)):

$$\text{TPC mg GAE/100 g FW} = \frac{\text{TPC per mL sample} \times \text{dilution factor} \times \text{total sample volume used}}{\text{Sample weight}} \times 100\% \quad (3)$$

### 2.8. Antioxidant Activity and Capacity

2.8.1. ABTS (2,2-Azino-bis(3-ethylbenzthiazoline-6-sulfonic acid)

The antioxidant activity of tomato fruit was measured using 2,2-azino-bis, 3-ethylbenzthiazoline-6-sulfonic acid according to the methods of Aadesariya and Pinheiro [20,21], with slight modification. ABTS was formed by reacting 7 mM ABTS aqueous solution with 2.45 mM of potassium per sulphate at $26 \pm 2\ °C$ and $60 \pm 5\%$ RH for 16 h in the dark. This solution was diluted in ethanol (around 1:89 $v/v$) before the test and equilibrated at $30\ °C$ to provide an absorbance of $0.700 \pm 0.02$ at 734 nm. The addition of 1 mL diluted ABTS solution in ethanol to 10 μL of sample extract was incubated at $30\ °C$ for 6 min before absorbance. The inhibition percentage for the blank absorbance was then calculated at 734 nm. The percentage of ABTS free radical inhibition was determined using the equation below (Equation (4)):

$$\text{ABTS inhibition (\%)} = \frac{(A0 - A1)}{A0} \times 100\% \quad (4)$$

where A0 = absorbance of the control and A1 = absorbance of sample

Solution A was prepared by dissolving 8 mg ABTS in 1 mL of water to obtain 7 mM ABTS solution. Solution B was prepared by dissolving 13.2 mg potassium per sulphate in 10 mL water to obtain 2.45 mM solution. Solution A (0.5 mL) was mixed with 0.5 mL of solution B and allowed to sit in darkness at $26 \pm 2\ °C$ for 12–16 h before use. The ABTS radical cation in this form is stable for 16 h.

2.8.2. Ferric Reducing Antioxidant Power

Tomato fruit tissue's antioxidant capacity was calculated using ferric reducing antioxidant power (FRAP). The assay was carried out according to the methods of Briones and Thaipong [22,23], with slight modifications. In FRAP assay, the FRAP reagent was freshly prepared by mixing 10 mM of 2,4,6-tris(2-pyridyl)-s-triazine (TPTZ) in 40 mM HCl solution, 300 mM acetate buffer ($C_2H_3NaO_2 \cdot 3H_2O$, pH 3.6), and 20 mM ferric chloride in the ratio of 1:10:1 ($v/v/v$). An aliquot of 50 μL sample extract was added to 950 μL FRAP reagent and incubated in a water bath of $37\ °C$ for 30 min. Absorbance was measured at 593 nm against a control that was prepared by adding 50 μL 80% methanol to 950 μL FRAP reagent. The standard curve was a linear line between 0 and 800 mM Trolox. The achieved results were expressed as μM Trolox equivalent (TE) of tomato fruit fresh weight using a standard curve with $R^2 = 0.98$. The obtained FRAP results were expressed in μM TE/g fresh weight and then calculated using the formula below (Equation (5)):

$$\text{FRAP μM TE/g FW} = \frac{\text{TE μM per mL} \times \text{dilution factor} \times \text{total sample volume used}}{\text{Sample weight}} \quad (5)$$

### 2.9. Determination of Defense Enzymes Activities

2.9.1. Protein Content

The extraction and analysis of protein were carried out using the combined techniques of Jumnongpon; Raseetha and Bonjoch [24–26], with minor modifications. The chemicals used to extract and evaluate enzymes were of analytical grade. A total of 0.5 g of frozen tomato fruit pulp tissue was immediately ground using a small ceramic kitchen pestle and mortar for 30 s on ice and homogenized with 1 mL ice-cold 50 mM phosphate buffer containing 1 M NaCl (pH 7.1). The mixture was centrifuged (Scan Speed 1730R, Scala

Scientific, the Netherlands) at 16,000× *g* at 4 °C for 20 min. The supernatant was then kept in an ice-water bath prior to the analysis.

The protein content of solutions derived from tomato fruit was measured using the Bradford procedure (Bradford 1976). The Bradford reagent was obtained from Bio-Rad Laboratories, Inc., Hercules, CA, USA. The reagent was prepared using distilled water in a 1:4 ratio; then, 1.2 mL of Bradford reagent was added with 120 μL protein supernatant and the mixture was briefly vortexed. The mixture was left to incubate for 30 min at room temperature, and the absorbance was read at 595 nm. The concentration of the extracted protein solutions from the bovine serum albumin standard curve ($R^2$ = 97) was quantified. The measurement was repeated three times. A standard curve plotting absorbance with various concentrations was obtained using bovine serum albumin (Sigma Chemicals Co., St. Louis, MO, USA) in the concentration range 25–400 μg/mL. The protein content in mg/mL was read against the standard curve and calculated using the formula specified by Wang [27] (Equation (6)):

$$\text{Protein content (mg/mL)} = \frac{\text{protein quality} \times \text{VT}}{\text{VS} \times \text{W}} \tag{6}$$

Protein quality results were collected in agreement with the standard curve; VT is the total volume of extraction, VS is the volume of solution for evaluation, and W is the weight of sample.

### 2.9.2. Determination of Phenylalanine Ammonia-Lyase (PAL) Enzyme Activity

The extraction for enzyme phenylalanine ammonia-lyase (PAL) was carried out according to Mohammed and Han [16,28], with some modifications. A total of 50 mg of frozen tissue was ground in 2 mL cold 25 mM sodium borate buffer (pH 8.8) containing 2 mM β-mercaptoethanol and 0.5 g polyvinylpyrrolidone. The homogenate was centrifuged (Scan Speed 1730R, Scala Scientific, Ede, The Netherlands) for 20 min at 16,000× *g* at 4 °C, and the supernatant was used as an enzyme source to determine the PAL activity.

PAL activity was determined by the production of cinnamate at 37 °C for 1 h; the absorbance was measured at 290 nm [29]. The assay mixture comprised 1 mL of enzyme extract and 2 mL of 50 mM sodium borate buffer (pH 8.8). The reaction started with 1 mL of 20 mM L-phenylalanine added and incubated at 37 °C for 1 h. Then, the reaction was stopped by adding 1 mL of 1 M HCl. The blank assay was performed with a mixture containing L-phenylalanine at zero incubation time. One unit of PAL activity was defined as the amount of enzyme that produced an absorbance increase of 0.01 at 290 nm per h [29]. The specific activity of the PAL enzyme was expressed as U/mg protein, where one unit of enzyme activity was defined as the production of cinnamic acid and the increase of one unit in absorbance per h. The activity of the enzyme was determined using the analytical approximation as defined in the following equation (Equation (7)):

Unit enzyme activity (U/mL) = ΔA 270 nm/min Test − ΔA 270 nm/min Blank × 3 × df/19.73 × 0.1　　　(7)

3 = total sample volume (mL)
df = dilution factor (weight/volume 50 mg/2 mL = 25)
19.73 = mM extinction coefficient of trans-cinnamate at 270 nm
unit definition: one unit will deaminate 1.0 μM of L-phenylalanine to trans-cinnamate and $NH_3$ per minute at pH 8.5 at 30 °C.

The specific activity of the enzymes was expressed in U/mg protein as followed: specific activity (U/mg protein) = unit activity (U/mL)/protein content (mg/mL) (Sigma Prod. No. P-2126).

### 2.9.3. Determination of Peroxidase Activity

Extraction and assay of peroxidase activity (POD) were carried out based on the combined procedure of Zhang and Raseetha [25,30], with minor modifications. A total of

0.5 g of frozen tomato fruit pulp tissue was immediately ground by using a small ceramic kitchen pestle and mortar for 30 s on ice and homogenized with 1 mL ice-cold 50 mM phosphate buffer containing 1 M NaCl (pH 7.1). The mixture was centrifuged for 20 min at 16,000× *g* at 4 °C. The supernatant was then kept in an ice-water bath prior to the analysis.

The POD activity was determined based on the development of brown coloration in the presence of $H_2O_2$, arising from the oxidation of guaiacol. A 20 µL sample extract supernatant was well mixed in a clean cuvette with 1.7 mL 0.1 M sodium phosphate buffer at pH 7.0 and 200 µL of 1 mM guaiacol. Then, the POD reaction was started by adding 100 µL 1.5% $H_2O_2$ *v/v*. The rate of absorbance rise at 485 nm was monitored for 3 min at 20 °C. The POD activity was expressed as U/mg protein by Kokkinakis and Ogola [31,32], as follows (Equation (8)):

$$\text{Unit activity (U/mL)} = (\Delta OD/\min \times V \times D)/(26.6 \times d \times v) \tag{8}$$

where $\Delta OD/\min$ = the increase in absorbance at 485 nm/min, V = total amount of reaction mixture (2 mL), D = enzyme dilution factor, 26.6 = mM extinction coefficient of guaiacol at 485 nm, d = light path length (cm) and v = volume of enzyme sample (0.02 mL)

The extinction coefficient was calculated using Beer-Lambert law ($\varepsilon = A/Lc$): $\varepsilon$ = extinction coefficient, A = absorption, L = path length (the thickness of the solution) and c = concentration of the solution.

The specific activity of the enzymes was expressed in U/mg protein, as follows (Equation (9)):

$$\text{Specific activity (U/mg protein)} = \text{Unit activity (U/mL)/Protein content (mg/mL)} \tag{9}$$

### 2.9.4. Determination of Polyphenol Oxidase Activity

Polyphenol oxidase (PPO) activity was determined based on changes in the color intensity of catechol oxidation, as described in the methods of Indunil and Mishra [33,34]. The extracted POD supernatant was used as the source of the enzyme, which was held at −20 °C. In brief, 200 µL of 0.01 M catechol was supplemented to start the reaction. The absorbance changes were recorded at 495 nm for 1 min. The PPO specific activity was determined by expressing PPO enzyme specific activity (U/mg protein) using the following equation (Equation (10)):

$$\text{Unit activity (U/mL)} = (\Delta OD/\min \times V \times D)/(11.3 \times d \times v) \tag{10}$$

where $\Delta OD/\min$ = the change in absorbance at 485 nm/min, V = total volume of reaction mixture (2.00 mL), D = enzyme dilution factor, 11.3 = mM extinction coefficient of catechol, d = light path length (1 cm) and v = volume of enzyme sample (0.2 mL)

The extinction coefficient was calculated by the Beer-Lambert law ($\varepsilon = A/Lc$): $\varepsilon$ = extinction coefficient, A = absorption, L = path length (the thickness of the solution) and c = concentration of the solution.

The specific activity of the enzymes was expressed in U/mg protein, as follows (Equation (11)):

$$\text{Specific activity (U/mg protein)} = \text{Unit activity (U/mL)/Protein content (mg/mL)} \tag{11}$$

### 2.10. Experimental Design and Statistical Analysis

The experiments were carried out using a completely randomized design (CRD), with eight coating treatments and four replications (Figure 2). The data obtained were analyzed using analysis of variance (ANOVA), and mean comparisons were performed using the least significant difference (LSD) at the significance level of $p \leq 0.05$. All the analyses were conducted using statistical analysis software (SAS) version 9.4 (SAS Institute Inc., Cary, NC, USA). The data in percent were transformed using a square root transformation before determining the significance level using LSD (Gomez and Gomez 1984). Pearson's

coefficients correlation were conducted to correlate the determined variables. The entire experiment was repeated four times, and the data were pooled before analysis. However, the positive control fruit could no longer be used for analysis after day 12 due to high disease severity and decay.

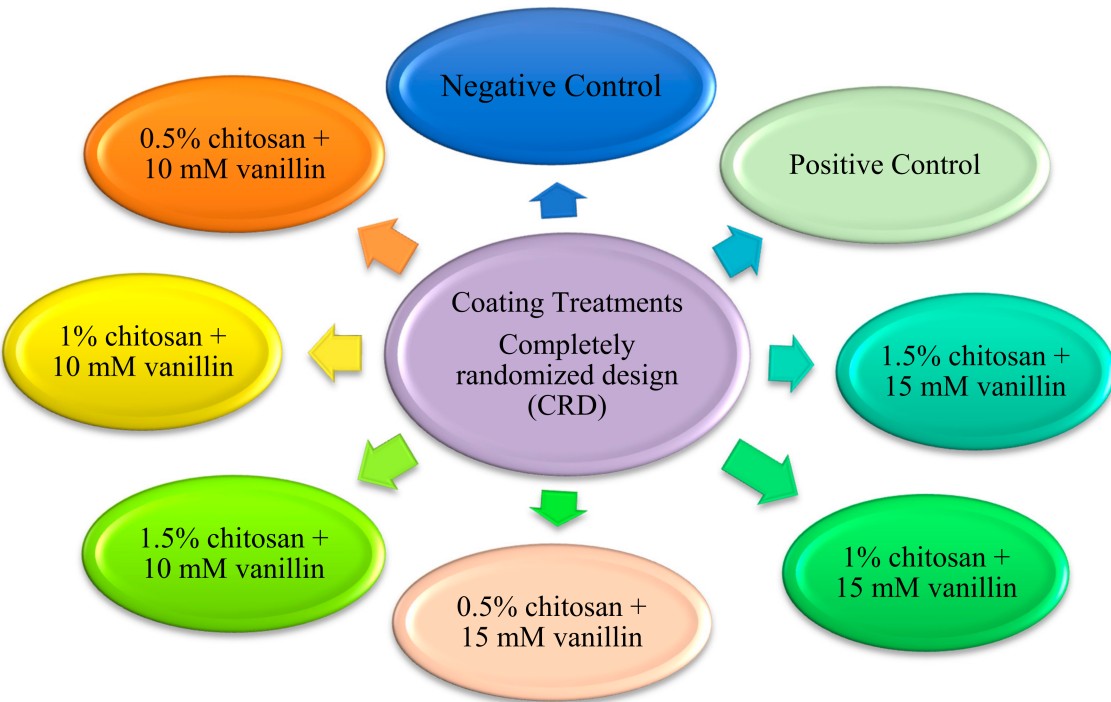

**Figure 2.** Schematic diagram of eight coating treatments used for *Fusarium oxysporum* inoculated tomato fruit stored for 15 days at 26 ± 2 °C and 60 ± 5% relative humidity.

## 3. Results

### 3.1. Disease Incidence and Diseases Severity

Table 2 shows that there was a significant interaction between coating treatments and storage day on the DS of tomato fruit.

Figure 3 shows that the incidence of the disease appeared after 6 days of storage, except the fruit treated with 1.5% chitosan + 15 mM vanillin. By storage day 9, DI in fruit treated with 1% chitosan + 15 mM vanillin and 1.5% chitosan + 15 mM vanillin remained low as compared to the negative and positive control fruit and those coated with 0.5% chitosan + 10 mM vanillin, 1% chitosan + 10 mM vanillin, 1.5% chitosan + 10 mM and 0.5% chitosan + 10 mM vanillin. This trend continued until storage day 15. At the end of storage, all fruit was severely infected by the disease, but the fruit treated with 1% chitosan + 15 mM vanillin and 1.5% chitosan + 15 mM vanillin showed lower incidence than other treatments.

Figure 4 shows that the DS appeared after 6 days of storage, except for the fruit treated with 1.5% chitosan + 15 mM vanillin. By storage day 9, disease severity in fruit treated with 1% chitosan + 15 mM vanillin and 1.5% chitosan + 15 mM vanillin was lower than the negative and positive control fruit and those coated with 0.5% chitosan + 10 mM vanillin, 1% chitosan + 10 mM vanillin, 1.5% chitosan + 10 mM vanillin and 0.5% chitosan + 15 mM vanillin. This trend continued until 15 days of storage. At the end of storage, all fruit were severely infected by the disease, but the fruit treated with 1% chitosan + 15 mM vanillin and 1.5% chitosan + 15 mM vanillin showed lower DS than other treatments.

From the Pearson's correlation analysis, there was strong significant positive correlation between disease incidence and severity (r = 0.94) (Table 3).

**Table 2.** Main and interaction effects of different coating treatments and storage days on disease incidence and severity of *Fusarium oxysporum* inoculated tomato fruit stored at $26 \pm 2\,^{\circ}C$ and $60 \pm 5\%$ relative humidity for 15 days.

| Factor | Disease Incidence (%) | Disease Severity (%) |
|---|---|---|
| Treatment | - | - |
| Negative control | 37.50 ab [z] | 40.83 ab |
| Positive control | 44.44 a | 50.83 a |
| 0.5% chitosan + 10 mM vanillin | 38.19 ab | 38.33 b |
| 1% chitosan + 10 mM vanillin | 31.25 b | 26.66 cd |
| 1.5% chitosan + 10 mM vanillin | 32.63 b | 22.50 d |
| 0.5% chitosan + 15 mM vanillin | 31.25 b | 34.16 bc |
| 1.0% chitosan + 15 mM vanillin | 9.02 c | 11.66 e |
| 1.5% chitosan + 15 mM vanillin | 6.94 c | 9.16 e |
| Storage day | - | - |
| 0 | 0.00 e | 0.00 e |
| 3 | 0.00 e | 0.00 e |
| 6 | 14.06 d | 16.87 d |
| 9 | 31.16 c | 36.25 c |
| 12 | 54.68 b | 51.25 b |
| 15 | 75.52 a | 71.25 a |
| Interaction | | |
| Treatment × Storage day | ** | ** |

[z] Mean values in a column followed by different letters indicate significant difference according to LSD at $p < 0.05$.
** Highly significant at $p \leq 0.05$ ($n = 24$).

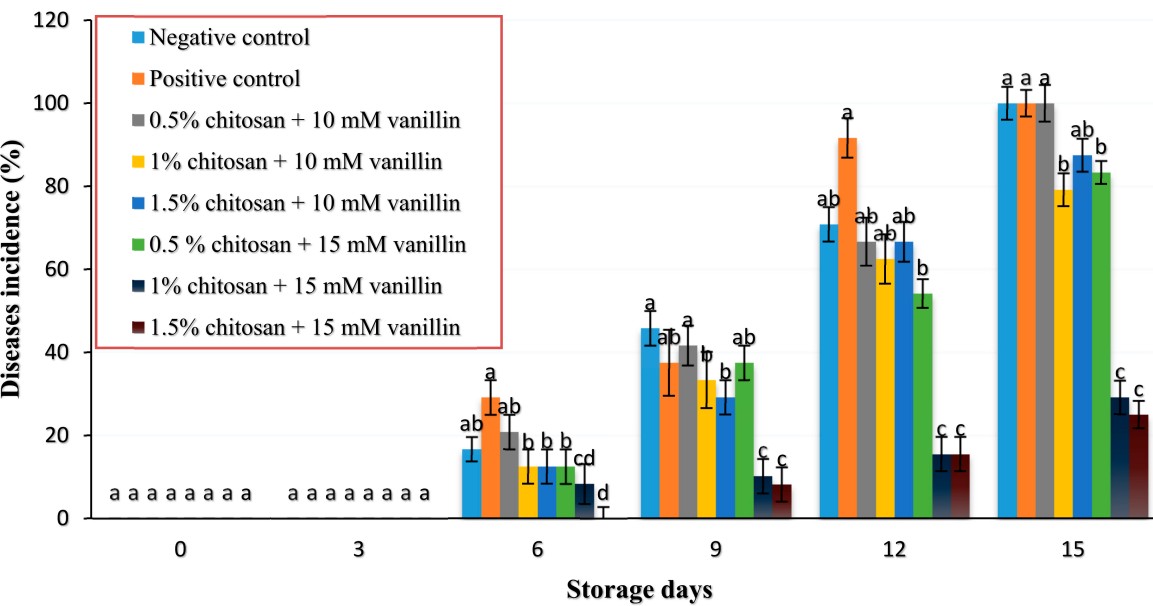

**Figure 3.** Effects of coating treatment on disease incidence of *Fusarium oxysporum* in tomato fruit stored for 15 days at $26 \pm 2\,^{\circ}C$ and $60 \pm 5\%$ relative humidity. Mean values in a column followed by different letters for each storage day differed significantly by LSD at $p \leq 0.05$. Vertical bars indicate standard error of means. Prior to analysis, the data were square root transformed, while non-transformed means are shown ($n = 24$).

### 3.2. Total Phenolic Content

There were significant interaction effects of treatment and storage day on the total phenolic content (TPC) of tomato fruit during storage (Table 4).

Figure 5 shows that coating treatment and storage day affected the total phenolic content of tomato fruit. By storage day 3, the fruit treated with 1% chitosan + 15 mM vanillin and 1.5% chitosan + 15 mM vanillin continued to have lower TPC than other treatments. However, at storage day 6, there was no significant difference among treatments. By storage

day 9, fruit treated with 1% chitosan + 15 mM vanillin and 1.5% chitosan + 15 mM vanillin obviously had lower TPC than positive and negative control fruit and those treated with 0.5% chitosan + 10 mM vanillin, 1% chitosan + 10 mM vanillin, 1.5% chitosan + 10 mM vanillin and 0.5% chitosan + 15 mM vanillin. This trend continued until the end of storage day 15.

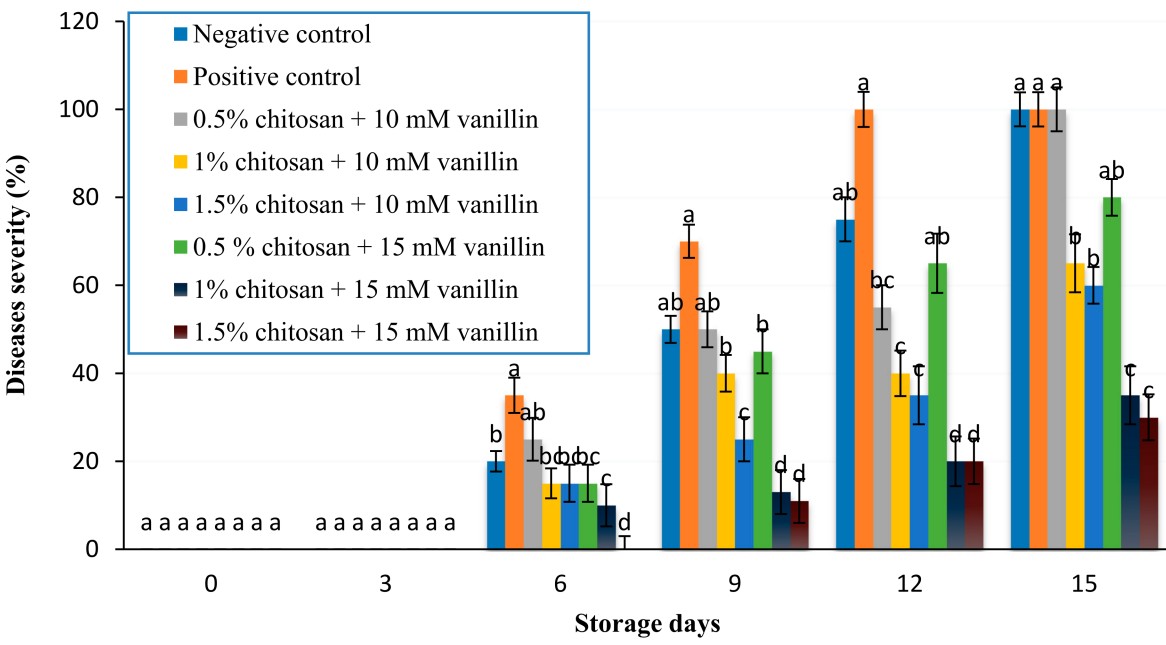

**Figure 4.** Effects of coating treatment on disease severity of *Fusarium oxysporum* in tomato fruit stored for 15 days at $26 \pm 2\,^{\circ}\text{C}$ and $60 \pm 5\%$ relative humidity. Mean values in a column followed by different letters for each storage day differed significantly by LSD at $p \leq 0.05$. Vertical bars indicate standard error of means. Prior to analysis, the data were square root transformed, while non-transformed means are shown (*n* = 24).

**Table 3.** Pearson's correlation coefficients for disease incidence and severity of *Fusarium oxysporum* inoculated tomato fruit stored at $26 \pm 2\,^{\circ}\text{C}$ and $60 \pm 5\%$ relative humidity for 15 days.

|  | Disease Incidence | Disease Severity |
|---|---|---|
| Disease incidence | - | - |
| Disease severity | 0.94 ** | - |

** Significant correlation at $p \leq 0.05$ and $p \leq 0.01$ (*n* = 24).

**Table 4.** Main and interaction effects of different coating treatments and storage days on antioxidant capacity of *Fusarium oxysporum* inoculated tomato fruit stored at $26 \pm 2\,^{\circ}\text{C}$ $60 \pm 5\%$ relative humidity for 15 days.

| Factor | Total Phenolic Content (mg GAE/100 g FW) | FRAP (mM TE/g FW) | ABTS (% Inhibition) |
|---|---|---|---|
| Treatment | - | - | - |
| Negative control | 48.08 a [z] | 1615.98 a | 37.99 a |
| Positive control | 51.61 a | 1562.32 a | 40.22 a |
| 0.5% chitosan + 10 mM vanillin | 48.31 a | 1639.26 a | 36.32 ab |
| 1% chitosan + 10 mM vanillin | 47.98 a | 1655.78 a | 36.81 ab |
| 1.5% chitosan + 10 mM vanillin | 43.86 b | 1418.87 b | 34.23 b |
| 0.5% chitosan + 15 mM vanillin | 46.38 ab | 1567.28 a | 36.69 ab |

**Table 4.** *Cont.*

| Factor | Total Phenolic Content (mg GAE/100 g FW) | FRAP (mM TE/g FW) | ABTS (% Inhibition) |
|---|---|---|---|
| 1.0% chitosan + 15 mM vanillin | 36.64 c | 1315.24 c | 29.82 c |
| 1.5% chitosan + 15 mM vanillin | 34.88 c | 1287.16 c | 28.60 c |
| Storage day | - | - | - |
| 0 | 33.65 c | 1703.27 b | 31.89 b |
| 3 | 34.11 bc | 1625.36 b | 29.14 c |
| 6 | 42.11 b | 1668.51 b | 33.15 ab |
| 9 | 44.51 a | 1256.06 c | 34.76 ab |
| 12 | 43.57 a | 1002.46 d | 37.48 a |
| 15 | 47.88 a | 1994.8 a | 39.80 a |
| Interaction Treatment × Storage day | ** | ** | * |

[z] Mean values in column followed by different letters indicate significant difference according to LSD at $p < 0.05$, * significant at $p \leq 0.05$, ** significant at $p \leq 0.05$, [ns] non-significant at $p \leq 0.05$ ($n = 24$).

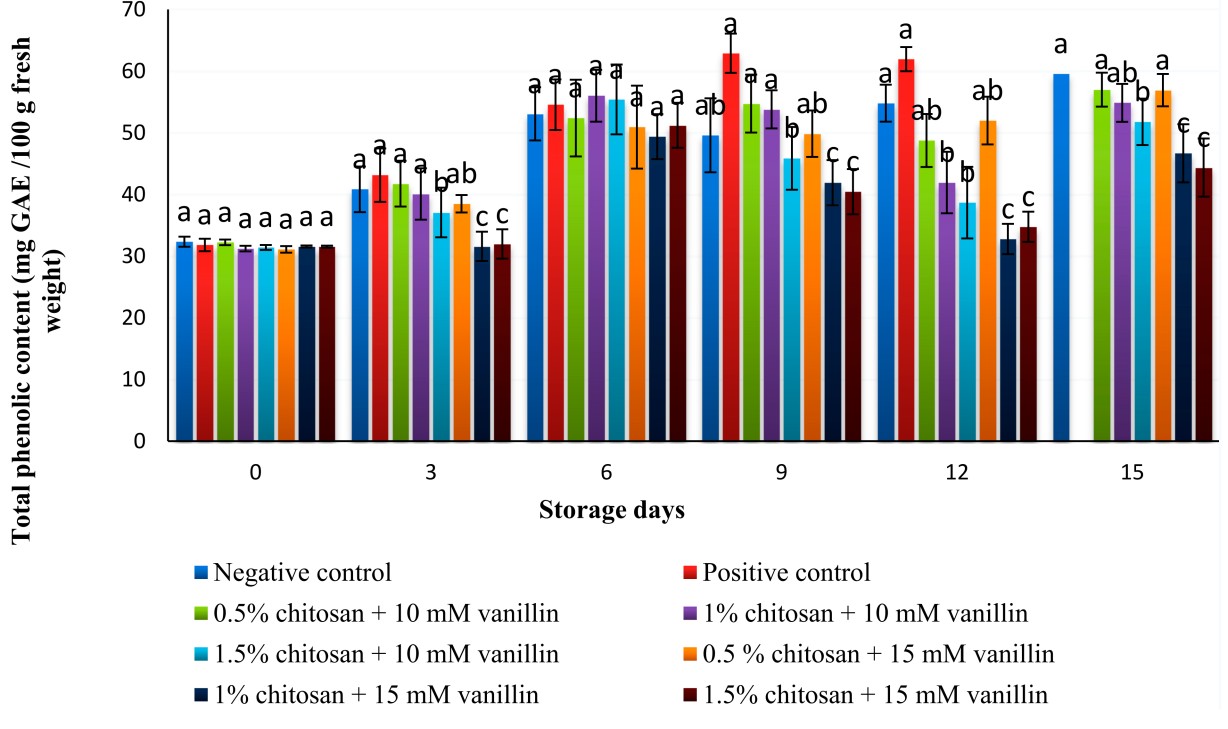

**Figure 5.** Effects of coating treatment on total phenolic content in *Fusarium oxysporum* inoculated tomato fruit stored for 15 days at 26 ± 2 °C and 60 ± 5% relative humidity. Mean values in a column followed by different letters for each storage day differed significantly by LSD at $p \leq 0.05$. Vertical bars indicate standard error of means ($n = 24$).

### 3.3. Antioxidant Capacity (FRAP and ABTS)

Table 4 indicates that there were highly significant interaction effects between coating treatments and storage duration of tomato fruit on the FRAP. At day 3, Figure 6 shows fruit coated with 1.5% chitosan + 10 mM vanillin, 1% chitosan + 15 mM vanillin and 1.5% chitosan + 15 mM vanillin had lower FRAP than positive and negative control fruit and also those coated with 0.5% chitosan + 10 mM vanillin, 1% chitosan + 10 mM vanillin and 0.5% chitosan + 15 mM vanillin. By day 6, fruit with 0.5% chitosan + 15 mM vanillin and 1% chitosan + 10 mM vanillin had greater FRAP than other fruit. Nevertheless, at day 9, there was no significant difference among treatment on fruit FRAP. This trend continued until day 12. By day 15 fruit coated with 1% chitosan + 15 mM vanillin and 1.5% chitosan + 15 mM vanillin had lower FRAP than negative control fruit and those coated

with 0.5% chitosan + 10 mM vanillin, 1% chitosan + 10 mM vanillin, 1.5% chitosan + 10 mM vanillin and 0.5% chitosan + 15 mM vanillin.

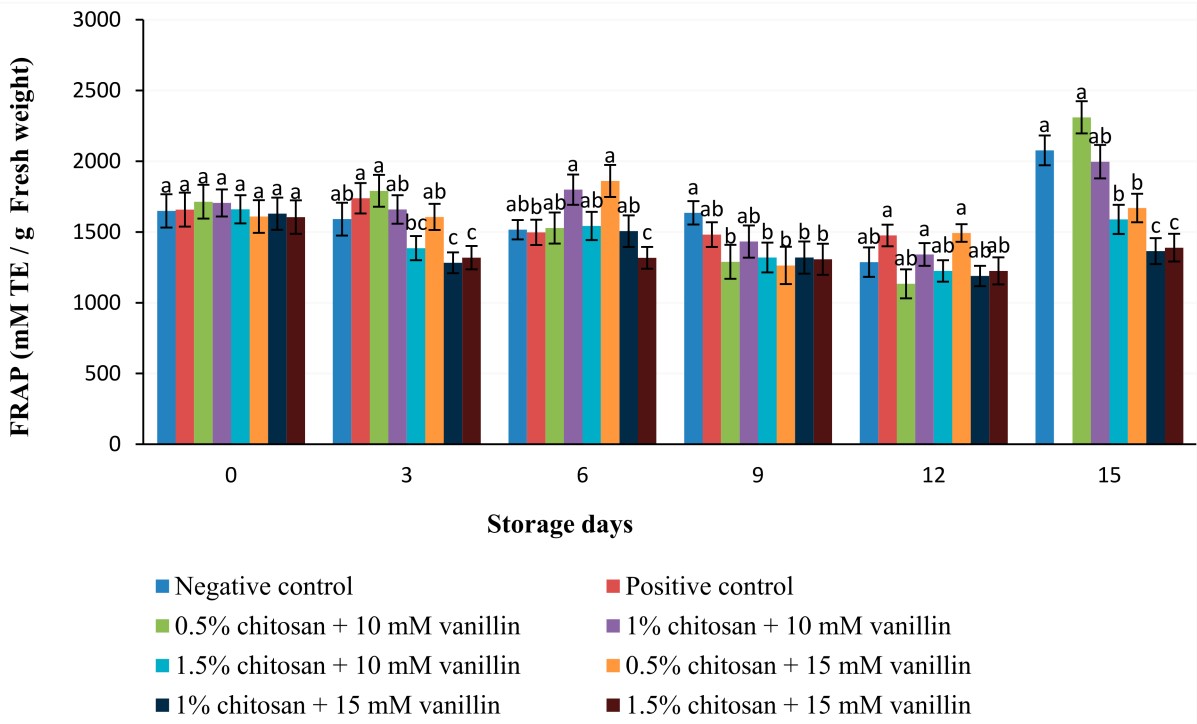

**Figure 6.** Effects of coating treatment on FRAP in *Fusarium oxysporum* inoculated tomato fruit stored for 15 days at 26 ± 2 °C and 60 ± 5% relative humidity. Mean values in a column followed by different letters in each storage day differed significantly by LSD at $p \leq 0.05$. Vertical bars indicate standard error of means ($n = 24$).

The interaction was significant between treatments and storage days in tomato fruit ABTS, 2, 2-azino-bis (3-ethylbenzthiazoline-6-sulfonic acid) (Table 4). According to the results as shown in Figure 7, there was no significant difference in ABTS among treatment at day 0. By storage day 3, fruit treated with 1.5% chitosan + 10 mM vanillin and 1.5% chitosan + 15 mM vanillin showed lower ABTS radical scavenging capacity than other treatments. At day 6, fruit coated with 1% chitosan + 15 mM vanillin and 1.5% chitosan + 15 mM vanillin had lower ABTS radical scavenging capacity than the control fruit and those treated with 0.5% chitosan + 10 mM vanillin, 1% chitosan + 10 mM vanillin, 1.5% chitosan + 10 mM vanillin and 0.5% chitosan + 15 mM vanillin. A similar trend was found in fruit stored for 9, 12 and 15 days.

There was a significant positive correlation between antioxidant (TPC) and antioxidant capacity (ABTS and FRAP) in tomato fruit treated with chitosan and vanillin during entire storage. From Pearson's correlation analysis, there was a significant positive correlation between TPC and ABTS (r = 0.53) and FRAP (r = 0.76). There was also a significant positive correlation between FRAP and ABTS (r = 0.62) (Table 5).

### 3.4. Defense-Related Enzyme (PAL, PPO and POD) Activity

In the present study, there were significant interaction effects between coating treatments and storage days in PAL, PPO and POD activities of tomato fruit (Table 6).

Figure 8 shows that there were no significant changes in PAL enzyme activity among treatments at day 0. At storage day 3, the activity of the enzyme dropped slightly in all fruit. However, by day 3, fruit treated with 1% chitosan + 15 mM vanillin and 1.5% chitosan + 15 mM vanillin showed lower PAL activity than positive and negative control fruit and also those coated with 0.5% chitosan + 10 mM vanillin, 1% chitosan + 10 mM vanillin, 1.5% chitosan + 10 mM vanillin and 0.5% chitosan + 15 mM vanillin. This trend continued until the end of storage day 15.

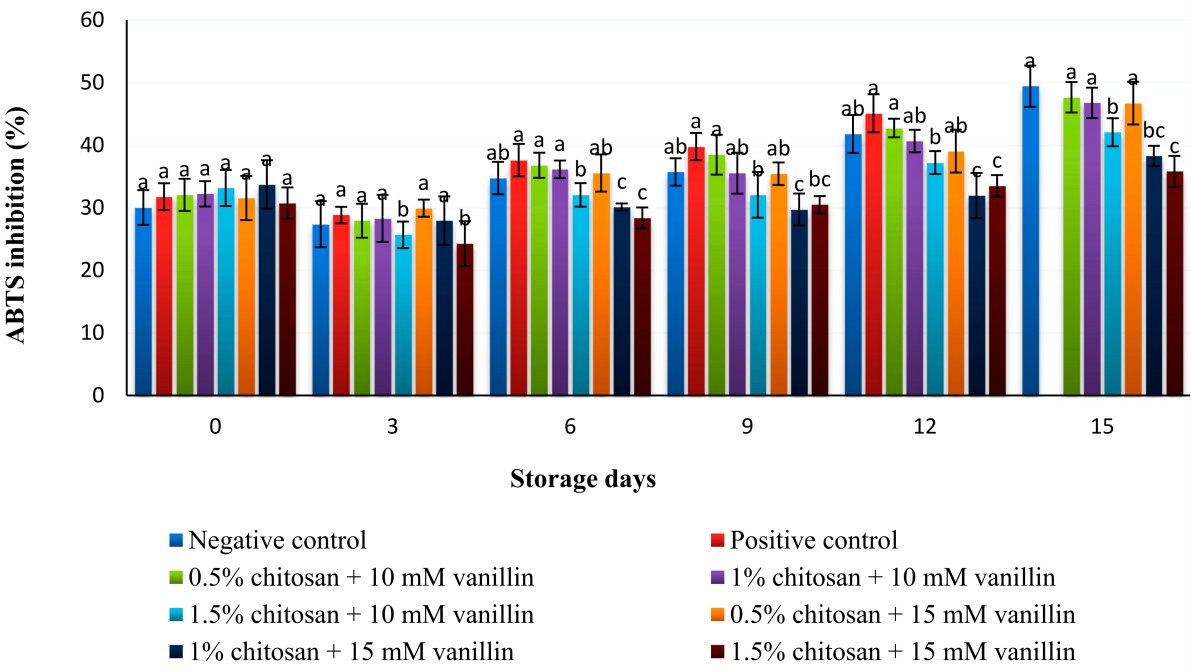

**Figure 7.** Effects of coating treatment on ABTS in *Fusarium oxysporum* inoculated tomato fruit stored for 15 days at 26 ± 2 °C and 60 ± 5% relative humidity. Mean values in a column followed by different letters for each storage day differed significantly by LSD at $p \leq 0.05$. Vertical bars indicate standard error of means ($n = 24$).

**Table 5.** Pearson's correlation coefficients for TPC, DPPH, ABTS and FRAP of *Fusarium oxysporum* inoculated tomato fruit stored at 26 ± 2 °C and 60 ± 5% relative humidity during 15 days of storage.

|  | TPC | ABTS | FRAP |
|---|---|---|---|
| TPC | - | - | - |
| ABTS | 0.53 ** | - | - |
| FRAP | 0.76 ** | 0.62 ** | - |

TPC = Total phenolic content, DPPH = 2,2-diphenyl-1-picrylhydrazyl, ABTS = 2,2′-azino-bis(3-ethylbenzothiazoline-6-sulphonic acid) and FRAP = ferric reducing antioxidant power. ** Significant correlation at $p \leq 0.05$ and $p \leq 0.01$ ($n = 24$).

**Table 6.** Main and interaction effects of different coating treatments and storage days on defense-related enzyme activity of *Fusarium oxysporum* inoculated tomato fruit stored at 26 ± 2 °C and 60 ± 5% relative humidity for 15 days.

| Factor | PAL Specific Activity (U/mg protein) | PPO Specific Activity (U/mg protein) | POD Specific Activity (U/mg protein) |
|---|---|---|---|
| Treatment | - | - | - |
| Negative control | 0.48 ab [z] | 0.64 b | 1.04 ab |
| Positive control | 0.53 a | 0.78 a | 1.10 a |
| 0.5% chitosan + 10 mM vanillin | 0.46 ab | 0.66 ab | 1.06 ab |
| 1% chitosan + 10 mM vanillin | 0.43 b | 0.58 c | 1.01 ab |
| 1.5% chitosan + 10 mM vanillin | 0.39 c | 0.64 b | 1.04 ab |
| 0.5% chitosan + 15 mM vanillin | 0.46 ab | 0.58 c | 0.96 bc |
| 1.0% chitosan + 15 mM vanillin | 0.35 d | 0.52 d | 0.86 c |
| 1.5% chitosan + 15 mM vanillin | 0.34 d | 0.54 d | 0.91 c |
| Storage day | - | - | - |
| 0 | 0.45 ab | 0.49 d | 0.74 d |
| 3 | 0.46 ab | 0.55 c | 1.05 b |
| 6 | 0.42 b | 0.52 cd | 0.87 c |
| 9 | 0.45 ab | 0.73 b | 0.93 bc |
| 12 | 0.48 a | 0.72 b | 1.24 a |

**Table 6.** *Cont.*

| 15 | 0.58 a | 0.84 a | 1.57 a |
|---|---|---|---|
| Interaction | ** | ** | ** |
| Treatment × Storage day | | | |

$^z$ Mean values in a column followed by different letters indicate significant difference according to LSD at $p < 0.05$. ** Highly significant at $p \leq 0.05$ ($n = 24$).

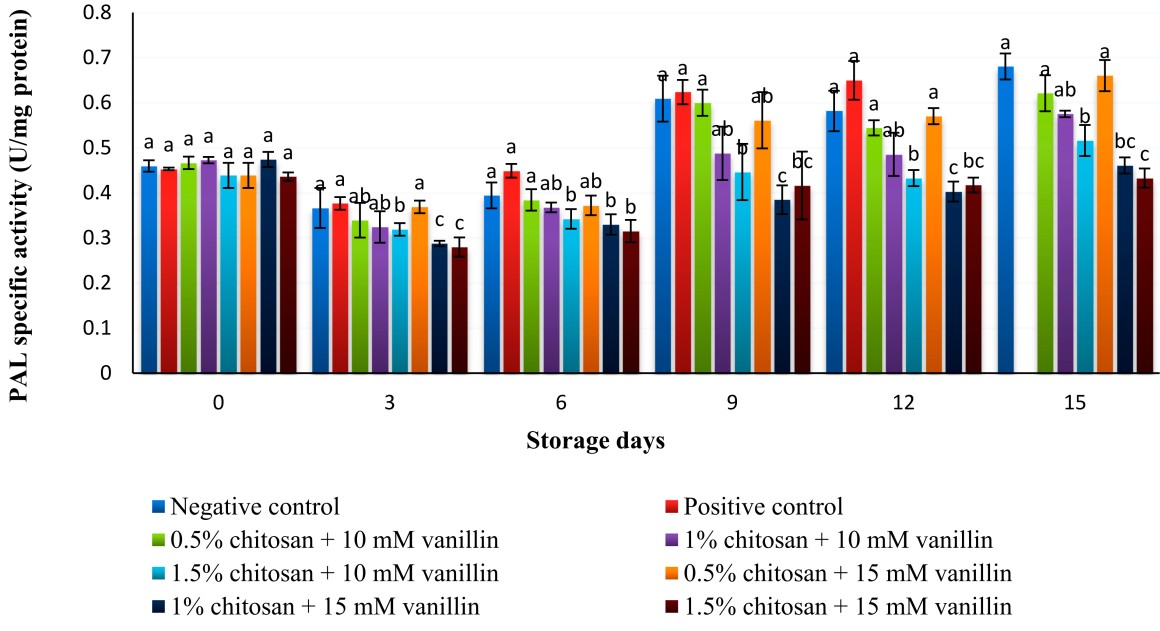

**Figure 8.** Effects of coating treatment on PAL specific activity in *Fusarium oxysporum* inoculated tomato fruit stored for 15 days at $26 \pm 2$ °C and $60 \pm 5\%$ relative humidity. Mean values in a column followed by different letters for each storage day differed significantly by LSD at $p \leq 0.05$. Vertical bars indicate standard error of means ($n = 24$).

Figure 9 shows that the PPO enzyme activity of tomato fruit at storage day 6 that was treated with 1% chitosan + 15 mM vanillin and 1.5% chitosan + 15 mM vanillin had lower PPO activity than positive and negative control fruit and also those coated with 0.5% chitosan + 10 mM vanillin, 1% chitosan + 10 mM vanillin, 1.5% chitosan + 10 mM vanillin and 0.5% chitosan + 15 mM vanillin. This trend continued until the end of storage day 15.

Figure 10 exhibits POD enzymes activity of tomato fruit increased slightly as storage day advanced to 3. Nevertheless, fruit treated with 1% chitosan + 15 mM vanillin and 1.5% chitosan + 15 mM vanillin shows lower POD activity than other treatments. By storage day 6, fruit coated with 1.5% chitosan + 10 mM vanillin, 1% chitosan + 15 mM vanillin and 1.5% chitosan + 15 mM vanillin shows lowest POD activity as compared to others treatment. This trend continued for the rest of storage day 15, however, the POD enzymes activity increased slightly in all treated fruit after 9 day of storage.

There was a significant correlation among defense-related enzymes. Pearson's correlation analysis shows that there was a highly significant positive correlation between PAL and PPO (r = 0.82), intermediate correlation of PAL and POD (r = 0.74) and intermediate correlation between POD and PPO (r = 0.67) (Table 7).

**Table 7.** Pearson's correlation coefficients for disease incidence and severity of *Fusarium oxysporum* inoculated tomato fruit stored at $26 \pm 2$ °C and $60 \pm 5\%$ relative humidity for 15 days.

| | PAL | PPO | POD |
|---|---|---|---|
| PAL | - | - | - |
| PPO | 0.82 ** | - | - |
| POD | 0.74 ** | 0.67 ** | - |

PAL = Phenylalanine ammonia-lyase, POD = Peroxidase and PPO = Polyphenoloxidase. ** Significant correlation at $p \leq 0.05$ and $p \leq 0.01$ ($n = 24$).

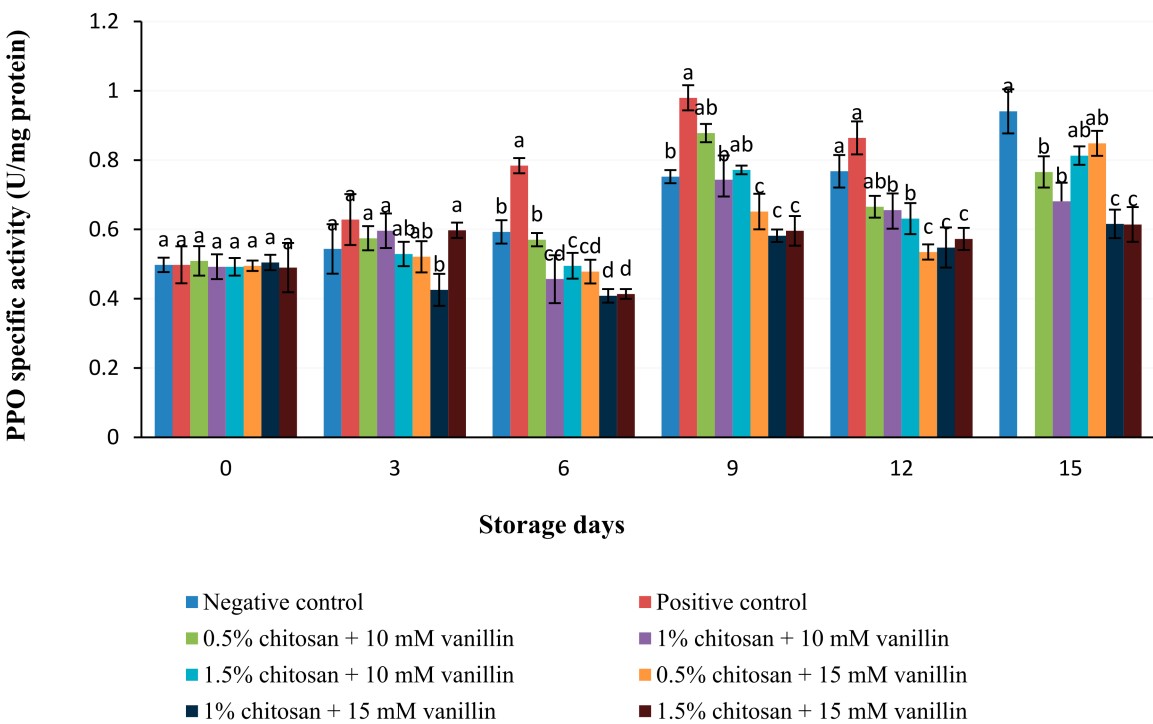

**Figure 9.** Effects of coating treatment on PPO specific activity in *Fusarium oxysporum* inoculated tomato fruit stored for 15 days at 26 ± 2 °C and 60 ± 5% relative humidity. Mean values in a column followed by different letters for each storage day differed significantly by LSD at $p \leq 0.05$. Vertical bars indicate standard error of means ($n = 24$).

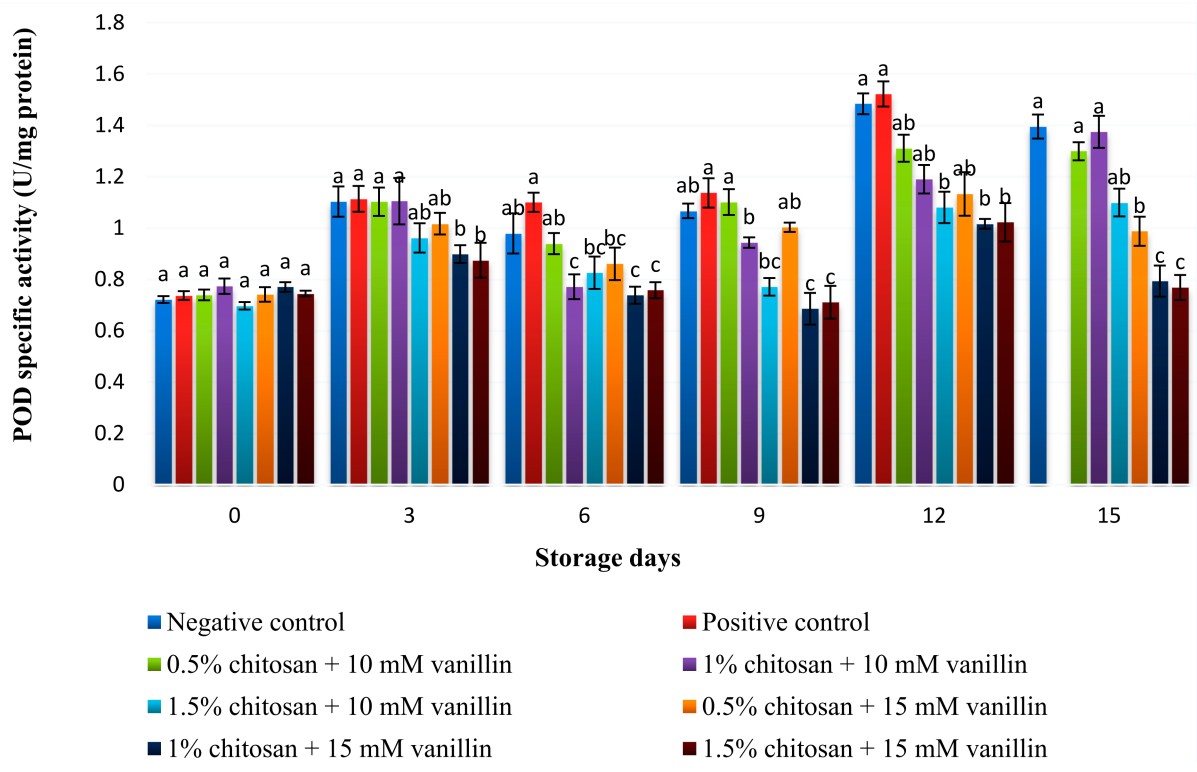

**Figure 10.** Effects of coating treatment on POD specific activity in *Fusarium oxysporum* inoculated tomato fruit stored for 15 days at 26 ± 2 °C and 60 ± 5% relative humidity. Mean values in a column followed by different letters for each storage day differed significantly by LSD at $p \leq 0.05$. Vertical bars indicate standard error of means ($n = 24$).

## 4. Discussion

### 4.1. Disease Incidence and Severity

The present study revealed that disease incidence and severity increased as the storage period advanced, while the coating significantly affected the percentage of disease incidence and severity during the storage period. As the storage days progressed, the control fruit and those treated with a low concentration of chitosan and vanillin showed more severe infection. In contrast, fruit coated with a high concentration of chitosan and vanillin had an inhibited progression of the disease in tomato fruit, as seen in Figures 3 and 4. Most probably, the chitosan coating formed a semi-permeable film around the fruit; this thin film could inhibit the growth of pathogens by disturbing the cell membrane of the pathogen that causes intracellular leakage and finally cell death. In addition, the chitosan coating can enhance the epidermal structure of fruit and limit the spread of the pathogens. Abebe and Mohammed [16,35] expressed that the coating could assist the cell wall in retaining its integrity against fungal attack and help in delaying pathogenic infection. This result was in agreement with the findings of Chen [36], where disease incidence and severity were lower in 1.5% chitosan-coated navel oranges than 0.5% chitosan when stored for 120 days at $5 \pm 0.5$ °C and 85–90% RH. Sikder [37] found that disease incidence and severity of bananas coated with 1% chitosan were lower as compared to 0.5% chitosan during 12 days storage at $28 \pm 2$ °C. In papayas, 1% chitosan significantly reduced anthracnose disease incidence and severity by 80% as compared to 0.05% chitosan when stored for 14 days at 13.5 °C and 96% RH [38]. In the present study, the barrier formed by the higher concentration of coating could have inhibited the growth of pathogens and slowed down the ripening and senescence process of tomato fruit, and therefore the disease incidence and severity were lesser in this fruit.

From the Pearson's correlation analysis, there was strong significant positive correlation between disease incidence and severity (r = 0.94) (Table 3). This was in agreement with Rashid [39], who found high correlation between disease incidence and severity (r = 0.91) in papaya fruit during 15 days of storage. In line with this study, Hossain [40] also found highly significant positive correlation between disease incidence and severity (r = 0.89) in banana fruit that was coated with 0.5, 0.75 and 1% chitosan and stored at $26 \pm 2$ °C and $85 \pm 5$% RH for 4 days. It is clear that disease incidence is a main contributor to disease severity.

### 4.2. Total Phenolic Content

Phenolic compounds, or secondary metabolites, are widely distributed in plants. They are particularly involved in plant defense against ultraviolet radiation and aggression by a pathogen [41]. Phenolic compounds are probably the most important candidates contributing to the antioxidant properties of plants and are associated with the scavenging of free radicals, breaking radical chain reactions, and chelating metals [42]. In this study, coating treatments had a significant effect on total phenolic content over the entire storage period. Their interaction effect was significant between treatment and storage day on the TPC of tomato fruit (Figure 5). In general, fruit treated with higher concentrations of chitosan and vanillin had lower TPC than those coated with low concentrations of chitosan and vanillin. However, Figure 3 illustrates that fruit treated with 1.5% chitosan + 15 mM vanillin had 25.5% lower TPC as compared to fruit coated with 0.5% chitosan + 1 mM vanillin at the end of storage day 15.

This lower TPC might be because the control fruit and those coated with low concentrations of chitosan ripened faster, and the phenolic compounds might have reacted with other compounds. It also appeared that the rise in phenolic levels may be due to biotic stresses, degradation of cells, and senescence [43]. In agreement with this study Munhuweyi [10] reported that pomegranate fruit coated with 1.5% chitosan had 36% lower total phenolic content when stored for 14 days at 4 °C as compared to fruit coated with 0.5% chitosan. In line with this study, previous researchers reported that a 1.5% chitosan coating caused a greater reduction in TPC as compared to fruit coated with 0.5% chitosan,

as found in sweet cherries [44], cut pineapple [45] and blueberries [46]. In the current study, the film created by the higher concentrations of coating slowed down the ripening and senescence process and suppressed abiotic stress in the fruit, modifying its metabolism and resulting in lower TPC.

### 4.3. Antioxidant Activity and Capacity

A number of assays have been introduced for the measurement of the total antioxidant activity of fruit [47]. In recent years, a wide range of spectrophotometric assays has been adopted to measure the antioxidant capacity of foods. The most popular are 2,2′-azino-bis-3-ethylbenzthiazoline-6-sulphonic acid (ABTS), 2, and ferric reducing antioxidant power (FRAP) [48]. Most of the assays employ the same principle: a synthetic colored radical or redox-active compound is generated, and the ability of a biological sample to scavenge the radical or to reduce the redox-active compound is monitored by a spectrophotometer.

Figure 6 shows that fruit coated with 1.5% chitosan + 15 mM vanillin had 30.45% lower FRAP than fruit coated with 0.5% chitosan + 10 mM vanillin at the end of storage day 15. The lower FRAP in tomato fruit coated with a higher concentration of chitosan and vanillin might be due to the formation of a protective barrier on the surface of fresh fruit that inhibits and reduces fruit antioxidant activity. Many researchers have reported that FRAP decreases during storage when fruit is coated using higher concentrations of chitosan, as occurred in pomegranates [49], strawberries [50] and tomatoes [51].

Figure 7 shows the ABTS of tomato fruit decreased as the concentration of chitosan and vanillin increased, while advancement of storage day increased its ABTS. Tomato fruit coated with 1.5% chitosan + 15 mM vanillin showed 19.66% lower ABTS than fruit coated with 0.5% chitosan + 10 mM vanillin. In line with this study, Martínez found that strawberry fruit coated with 1.5% chitosan had lower ABTS than fruit coated [52] with 0.5% chitosan during 15 days of storage. Most probably, the barrier formed by the higher concentration of coating delayed the senescence process and reduced decay in tomato fruit, and thus ABTS was lower in this fruit.

From Pearson's correlation analysis, there was a significant positive correlation between TPC and ABTS (r = 0.53) and FRAP (r = 0.76). There was also a significant positive correlation between FRAP and ABTS (r = 0.62) (Table 5). This was in agreement with Sushant et al. (2019) who found highly positive correlation between TPC and DPPH (r = 0.75) in Cassia tora plant. Similar result was also reported by Floegel et al. (2011) that there was highly significant in apple fruit that the correlation between TPC and DPPH (r = 0.89), highly significant correlation TPC and ABTS (r = 0.94) and strong significant correlation of TPC and FRAP (r = 0.70). In line with this study, Fu et al. (2010) also found highly significant positive correlation between TPC and antioxidant capacity (FRAP) (r = 0.79) in *Ficus benjamina*. The finding of this study indicated that TPC is the major contributor for tomato fruit antioxidant capacity.

### 4.4. Effects of Coating on the Activity of Defense-Related Enzymes (PAL, PPO and POD)

PAL, PPO and POD are among the most important enzymes having defensive responses in plants against insects and pathogens [53]. PAL is a key enzyme in the metabolism of phenols that protect plants against stress conditions [54]. There was a significant interaction effect between treatment and storage day on tomato fruit defensive enzyme PAL activity (Figure 8). At the end of storage day 15, PAL activity of fruit coated with 1.5% chitosan + 15 mM vanillin was 44.18% lower than those coated with 0.5% chitosan + 10 mM vanillin. A study by Zhan and Zhu [55] found that the PAL activity of water caltrop fresh fruit (*Trapa natans* L.) coated with 1% and 2% chitosan was lower than those coated with 0.5% chitosan during 15 days of storage at 4 ± 1 °C and 80%–85% RH. Previous researchers also reported that jujube fruit (*Ziziphus jujuba* Mill.) [56] and tomato fruit [57] with 1.5% chitosan coating had lower PAL activity than those coated with 0.5%. In the present study, the layer created by the higher concentration of coating most probably reduced ethylene

production rate and thus slowed down the ripening and senescence process of tomato fruit, leading to low PAL activity.

PPO is a key defense enzyme against pathogen reaction through the oxidation of polyphenols into quinines, which have antimicrobial activity and also strengthen the resistance of plant cells during microbial attack [27,58]. Figure 9 shows that, at storage day 15, the PPO activity of tomato fruit coated with 1.5% chitosan + 15 mM vanillin was 21.4% lower than the PPO activity of fruit coated with 0.5% chitosan + 10 mM vanillin. In agreement with this study, Minh [59] found the PPO activity of fresh mushrooms coated in 1.5% chitosan was lower than those coated with 0.5% chitosan. A study by Ghasemnezhad [49] demonstrated that PPO activity in pomegranate fruit coated with 1% chitosan was lower than those coated with 0.5% chitosan. A similar finding was also reported in litchi fruit [60] and tomato fruit [61] during storage. The reduction of PPO activity in high concentration chitosan coated tomato fruit might be due to low respiration and ethylene production rates, reducing disease attack and slowing ripening and senescence processes.

POD is one of the enzymes expressed in different stimuli, including pathogenic challenges, and has important roles in pathogenesis, oxidative burst, and resistance to infection [62]. As the storage day of tomato fruit advanced, fruit POD activity increased; in contrast, as the concentration of chitosan and vanillin coating increased, the POD activity decreased (Figure 10). However, fruit coated with 1.5% chitosan + 15 mM vanillin had 40.9% lower POD activity than fruit coated with 0.5% chitosan + 10 mM vanillin at the end of storage day 15. In line with this study, Ismail [63] found that fresh green beans coated with 1.5% chitosan had lower POD than those coated with 0.5% chitosan stored at 4 °C and 85%–90% RH for 28 days. In agreement with this study, previous researchers reported that 1.5% chitosan had lower POD in fruit than those coated with 0.5% chitosan, as found in tomato fruit [5], mushrooms [60] and strawberries [50]. In the current study, the film formed by the higher concentration of coating reduced disease attack and cell structure damaged by the pathogen and also slowed down respiration rate, ripening and senescence processes of tomato fruit; thus POD activity was lower in this fruit.

Pearson's correlation analysis shows that there was a highly significant positive correlation between PAL and PPO ($r = 0.82$), intermediate correlation of PAL and POD ($r = 0.74$) and intermediate correlation between POD and PPO ($r = 0.67$) (Table 7). The result was in agreement with Adiletta [64], who found higher correlation between PPO and POD ($r = 0.79$) in loquat fruit coated with 1% chitosan and stored at 7 °C for 21 days. In line with this study, Pasquariello [65] also found a highly positive correlation between PPO and POD ($r = 0.87$) and PPO and PAL ($r = 0.71$) in strawberry fruit coated with 1% chitosan stored at 2 °C and 95% RH for 14 days. This result indicated that defense-related enzymes such as PAL, PPO and POD are the main contributor to the oxidation of polyphenols into quinines, which strengthen the resistance of the plant cells during microbial attack.

## 5. Conclusions

The chitosan and vanillin coating could be considered as a commercial application to improve shelf life and maintain tomato fruit quality during storage at a room temperature of 26 ± 2 °C and at 60 ± 5% RH. The present findings show that chitosan and vanillin coating can effectively inhibit postharvest diseases in tomato fruit by controlling the disease incidence and severity as well as by keeping constant the defense-related enzyme activity. Furthermore, chitosan and vanillin consistently maintain the antioxidant activity and capacity. Our results suggest that a chitosan and vanillin coating of 1.5% chitosan + 15 mM vanillin formed a protective layer on fruit surfaces that helped to inhibit disease occurrence, slowing down the ripening and senescence processes in tomato fruit. As a result, tomato fruit effectively stored under 26 ± 2 °C and 60 ± 5% RH for 15 days, even inoculated with *Fusarium oxysporum*.

**Author Contributions:** Data curation, Z.S.S. and S.F.Y.; formal analysis, Z.S.S. and S.F.Y.; funding acquisition, P.D.; investigation, Z.S.S.; methodology, Z.S.S.; project administration, P.D. and J.J.N.;

resources, Z.S.S. and P.D.; supervision, P.D.; writing—original draft, Z.S.S.; writing—review & editing, P.D., J.J.N. and S.F.Y. All authors have read and agreed to the published version of the manuscript.

**Funding:** This research received no external funding. The research was supported by supervisor Phebe Ding and the publication fee was funded by UPM Research Management Center.

**Institutional Review Board Statement:** Not applicable.

**Informed Consent Statement:** Not applicable.

**Data Availability Statement:** The data used to support the findings of this study are included within the article.

**Conflicts of Interest:** The authors declare that they have no known competing financial interests or personal relationships that could have appeared to influence the work reported in this paper.

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
