# Peer review of "Controlling Fusarium oxysporum Tomato Fruit Rot under Tropical Condition Using Both Chitosan and Vanillin"

_coatings, doi:10.3390/coatings11030367_

Round 1

Reviewer 1 Report

The manuscript contains very important issues regarding the evaluation of the antimicrobial activity of chitosan and vanillin against the inoculated pathogen and the mitigation of the effect of chitosan and vanillin coating in vivo on Fusarium oxysporum fruit rot and defense of related enzymes PAL, PPO and POD. Chitosan and vanillin in aqueous solutions, from 0.5 to 1.5% chitosan + 10 or 15 mM vanillin, used as an edible coating on tomatoes and stored at 26°C / 60 relative humidity. The authors proved that chitosan and vanillin coating can effectively inhibit postharvest diseases of tomato fruit by controlling the disease incidence and severity as well as keeping constant defense-related enzymes activity. It is also important that such a coating maintain the antioxidant activity and capacity of the fruit. Both inhibiting the development of diseases, slowing down the ripening and aging process of tomato fruits proved to be very effective in storing tomato fruits.

All parts of the manuscript, introduction, methodology experimental procedures, results and conclusions were presented properly. The manuscript covers a wide research scope, the results are correctly described and presented in tables and figures, including statistical analysis, and were discussed extensively with the available literature. These references correctly applied. Many sources (65 items) were used, mainly from recent years. The conclusions are supported by the evidence.

I suggest checking if it is correctly written in figure captions: "Vertical bars indicate standard error of means for four replicates. (N = 24)."

Only References remains to be corrected, literature items are not entered equally, e.g. year: "(2013)" or "2001.", publication titles are written in lowercase (as in a sentence) or in capital letters.

The text is written in understandable language, grammatically correct, there are a few minor errors, such as unnecessary spaces (e.g. line 631).

The research has a scientific dimension and can be widely used in practice. The presented test results with a chitosan and vanillin coating can be considered as a commercial application to extend the shelf life and maintain the quality of tomato fruit during storage at room temperature.

I recommend publishing this manuscript. Congratulations to the authors

Author Response

The manuscript contains very important issues regarding the evaluation of the antimicrobial activity of chitosan and vanillin against the inoculated pathogen and the mitigation of the effect of chitosan and vanillin coating in vivo on Fusarium oxysporum fruit rot and defense of related enzymes PAL, PPO and POD. Chitosan and vanillin in aqueous solutions, from 0.5 to 1.5% chitosan + 10 or 15 mM vanillin, used as an edible coating on tomatoes and stored at 26°C / 60 relative humidity. The authors proved that chitosan and vanillin coating can effectively inhibit postharvest diseases of tomato fruit by controlling the disease incidence and severity as well as keeping constant defense-related enzymes activity. It is also important that such a coating maintain the antioxidant activity and capacity of the fruit. Both inhibiting the development of diseases, slowing down the ripening and aging process of tomato fruits proved to be very effective in storing tomato fruits.

All parts of the manuscript, introduction, methodology experimental procedures, results and conclusions were presented properly. The manuscript covers a wide research scope, the results are correctly described and presented in tables and figures, including statistical analysis, and were discussed extensively with the available literature. These references correctly applied. Many sources (65 items) were used, mainly from recent years. The conclusions are supported by the evidence.

I suggest checking if it is correctly written in figure captions: "Vertical bars indicate standard error of means for four replicates. (N = 24)."

Answer 1; Dear respectful reviewer thanks for your kind comments. It has been revised.

Only References remains to be corrected, literature items are not entered equally, e.g. year: "(2013)" or "2001.", publication titles are written in lowercase (as in a sentence) or in capital letters.

Answer 2: It has been revised.

The text is written in understandable language, grammatically correct, there are a few minor errors, such as unnecessary spaces (e.g. line 631).

 Answer 3: It has been revised.

The research has a scientific dimension and can be widely used in practice. The presented test results with a chitosan and vanillin coating can be considered as a commercial application to extend the shelf life and maintain the quality of tomato fruit during storage at room temperature.

I recommend publishing this manuscript. Congratulations to the authors

Reviewer 2 Report

The research in this paper is well conducted and comprehensive. The results are scientifically sound and important for different crops including tomatoes. However I think that is necesary to include comparisons with other work with this subject.

Author Response

Comments and Suggestions for Authors

The research in this paper is well conducted and comprehensive. The results are scientifically sound and important for different crops including tomatoes. However, I think that is necessary to include comparisons with other work with this subject.

Answer;

Dear respectful reviewer thanks for your kind comments and suggestion and your encouragement. The compression was carried out in discussion part. Now I don’t have access to lab and fungi to use in other crops as well and then compared. I will try to use more compression in discussion part, tomato fruit with other crops as per literature.

Reviewer 3 Report

The authors have conducted plenty of experiments, howerver is similar to their previous work (also published in coatings)

Major comments

If I am not wrong, the work is a continuation/ or similar to their previous work “ Combining Chitosan and Vanillin to Retain Postharvest Quality of Tomato Fruit during Ambient Temperature Storage. Coatings 2020, 10(12), 1222.

I would like to know what is the major difference (just tropical and ambient temeprature) between the two works and how this is different. Authors recommend 1.5% chitosan + 15 mM vanillin in both cases.  

Have the authors tried with Chitosan and vanillin alone?

What is the rationale for combining these two of them? Is it economical?

Minor comments

Writing can be modified.

Fig. 1. Figure legend should be elaborated and made self-explanatory.

Fig. 2 also needs some explanation.

Author Response

Major comments

If I am not wrong, the work is a continuation/ or similar to their previous work “Combining Chitosan and Vanillin to Retain Postharvest Quality of Tomato Fruit during Ambient Temperature Storage. Coatings 2020, 10(12), 1222.

I would like to know what is the major difference (just tropical and ambient temeprature) between the two works and how this is different. Authors recommend 1.5% chitosan + 15 mM vanillin in both cases.  

Have the authors tried with Chitosan and vanillin alone?

Answer 1:

Dear respectful reviewer thanks for your kind comments. This two experiments and works actually are not same. As you mentioned in my previous works, I have evaluated effect of chitosan and vanillin combination on tomato fruit post-harvest quality and general disease incidence. I have not inoculated any fungi to tomato. I have stored in tropical conditions for 25 days and each evaluation was carried out in 5 days’ interval. In previous works I have did not evaluated antioxidant properties and defense related enzymes. In previous experiment I have seven 7 treatments

In this work I have identified fusarium oxysporum as costive agent of tomato fruit rot in tropical condition. So I have inoculated the mentioned fungi to tomato fruits and evaluated in 3 days’ interval, and evaluated the effects of same coating agent on antioxidant, antioxidant properties, defense related enzyme as well as Fusarium oxysporum disease incidence and severity. I have did not evaluated postharvest quality such is TSS, weight loss, ph etc… In this experiment I have 8 treatments.

What is the rationale for combining these two of them? Is it economical?

Answer 2:

As per literature and mine preliminary study in vivo chitosan is week antimicrobial agent but has very effective properties in preserving post-harvest quality, antioxidant and antioxidant properties and could keep defense related enzymes. However, vanillin has strong antimicrobial properties, I have combined both coating agent chitosan and vanillin to achieve both goals keep tomato antioxidant, antioxidant properties, defense related enzymes as well as control disease incidence and severity.

My preliminary study link

https://www.researchgate.net/profile/Zahir-Shah-Safari/publication/349494597_Antifungal_Evaluation_Of_Edible_Coating_Agent_Against_Fusarium_Oxysporum_On_Tomato/links/6033930892851c4ed58abf09/Antifungal-Evaluation-Of-Edible-Coating-Agent-Against-Fusarium-Oxysporum-On-Tomato.pdf

Minor comments

Writing can be modified.

Fig. 1. Figure legend should be elaborated and made self-explanatory.

Answer 3:

Revised

Fig. 2 also needs some explanation.

 Answer 4:

Revised

Round 2

Reviewer 3 Report

The comments are addressed. Thank you